# Induced Models of Osteoarthritis in Animal Models: A Systematic Review

**DOI:** 10.3390/biology12020283

**Published:** 2023-02-10

**Authors:** Umile Giuseppe Longo, Rocco Papalia, Sergio De Salvatore, Riccardo Picozzi, Antonio Sarubbi, Vincenzo Denaro

**Affiliations:** 1Research Unit of Orthopaedic and Trauma Surgery, Fondazione Policlinico Universitario Campus Bio-Medico, Via Alvaro del Portillo 200, 00128 Roma, Italy; 2Research Unit of Orthopaedic and Trauma Surgery, Department of Medicine and Surgery, Università Campus Bio-Medico di Roma, Via Alvaro del Portillo 21, 00128 Roma, Italy; 3Department of Orthopedics, Children’s Hospital Bambino Gesù, 00165 Roma, Italy

**Keywords:** osteoarthritis, animal model, pig, rats, mice, induction method

## Abstract

**Simple Summary:**

This study aimed to determine the most appropriate animal model for the study of osteoarthritis (OA). OA is a common joint disease and can be induced in animal models through mechanical, surgical, or chemical methods. The study conducted a literature review of articles from January 2010 to July 2021 on various databases such as Medline, Embase, and Google Scholar. After reviewing 1621 articles, 36 studies were selected that included a total of 1472 animals. The results showed that various induction models, including mechanical, surgical, and chemical, have been proven to be suitable for inducing OA in animals. However, there is still not a clear "gold standard" for choosing the best animal model for OA. This study provides valuable information for researchers to make informed decisions about the appropriate animal model for their specific OA research questions. The findings of this study will be valuable to society by helping to advance our understanding of OA and develop better treatments for this debilitating disease.

**Abstract:**

The most common induction methods for OA are mechanical, surgical and chemical. However, there is not a gold standard in the choice of OA animal models, as different animals and induction methods are helpful in different contexts. Reporting the latest evidence and results in the literature could help researchers worldwide to define the most appropriate indication for OA animal-model development. This review aims to better define the most appropriate animal model for various OA conditions. The research was conducted on the following literature databases: Medline, Embase, Cinahl, Scopus, Web of Science and Google Scholar. Studies reporting cases of OA in animal models and their induction from January 2010 to July 2021 were included in the study and reviewed by two authors. The literature search retrieved 1621 articles, of which 36 met the selection criteria and were included in this review. The selected studies included 1472 animals. Of all the studies selected, 8 included information about the chemical induction of OA, 19 were focused on mechanical induction, and 9 on surgical induction. Nevertheless, it is noteworthy that several induction models, mechanical, surgical and chemical, have been proven suitable for the induction of OA in animals.

## 1. Introduction

Osteoarthritis (OA) is the most prevalent form of arthritis among middle-aged and elderly individuals [1]. The etiology of OA is multi-factorial, and the underlying pathogenic mechanisms are not fully understood. While the progression of the disease is typically slow, it can result in significant pain and disability, negatively impacting the quality of life. Studying OA in humans can be difficult, hence animal models have been utilized to assess OA development and evaluate novel treatments.

Several animal models have been developed for the study of OA, but it can be challenging to determine the most suitable one for specific research purposes [2]. Ideal experimental models should accurately mimic the disease and enable the joint analysis of relevant tissues using biomechanical, radiographic and microscopic evaluations. Common surgical models of knee OA include the anterior cruciate ligament transection model (ACLT), the partial medial meniscectomy model or a combination of both [3]. Other examples include rat ACLT and partial meniscectomy in rabbits. A widely studied and well-evaluated mouse model involves destabilizing the medial meniscus by surgically transecting the tibial ligament. To examine the potential pathogenic mechanisms leading to morphological changes, researchers have analyzed the morphological responses of the articular cartilage and subchondral bone after surgical instability, mechanical unloading and intra-articular injection [4]. Induction methods for OA include mechanical, surgical and chemical methods; however, there is no gold standard for choosing OA animal models, as different animals and induction methods are useful in different situations.

The aim of this review is to clearly define the most reliable animal model for different OA conditions by presenting the newest evidence and outcomes found in the literature and using a systematic approach. This could assist researchers worldwide in determining the most accurate indication for developing an OA animal model, discover new ways to study this condition and test pharmaceuticals or surgical techniques to combat the disease.

## 2. Materials and Methods

### 2.1. Study Selection

The PIOS approach (Patient (P), Intervention (I), Outcome (O) and Study Design (S)) was used to formulate the research question. This research aimed to report the most reliable animal models (P) and their induction methods (I) to better study OA development (O) in different contexts.

### 2.2. Inclusion Criteria

Only studies that reported cases of OA in animal models made through mechanical, surgical and chemical inductions were included.

Only studies regarding knee, hip, shoulder and spine were included. Only English-language articles were considered. The Oxford categorization was used to filter peer-reviewed publications at each level of evidence.

### 2.3. Exclusion Criteria

Studies focusing on spontaneous OA animal models were excluded, as were studies regarding human models of OA and other forms of arthritis. Studies that selected genetic manipulations as a method of induction of OA in animal models were also discarded. Only the most recent literature was evaluated; therefore, studies conducted prior to 2010 were also excluded. Technical notes, letters to editors and instructional courses were excluded. Studies with a sample size of less than ten animals were deemed ineligible for the current study. Cases involving temporomandibular joint OA were also excluded. Studies with specific therapeutic purposes and drug tests were excluded, specifically, OA inductions following diets, oral administration of drugs and biological alterations.

### 2.4. Search

The systematic review was carried out using the Preferred Reporting Items for Systematic Reviews and Meta-analyses (PRISMA) criteria. Bibliographic databases Medline, Embase, Cinahl, Scopus, Web of Science and Google Scholar were searched. The following was the string-searching used: ((“models, animal”[MeSH Terms] OR (“models”[All Fields] AND “animal”[All Fields]) OR “animal models”[All Fields] OR (“animal”[All Fields] AND “model”[All Fields]) OR “animal model”[All Fields]) AND (“osteoarthritis”[MeSH Terms] OR “osteoarthritis”[All Fields] OR “osteoarthritides”[All Fields]) AND (“chemical”[All Fields] OR “chemicals”[All Fields] OR “chemically”[All Fields] OR “chemicals”[All Fields]) OR “surgical procedures, operative”[MeSH Terms] OR (“surgical”[All Fields] AND “procedures”[All Fields] AND “operative”[All Fields]) OR “operative surgical procedures”[All Fields] OR “surgical”[All Fields] OR “surgically”[All Fields] OR “surgicals”[All Fields] OR (“mechanical”[All Fields] OR “mechanically”[All Fields] OR “mechanicals”[All Fields] OR “mechanics”[MeSH Terms] OR “mechanics”[All Fields] OR “mechanic”[All Fields])) NOT (“mesenchym”[All Fields] OR “mesenchymal”[All Fields] OR “mesenchymalized”[All Fields] OR “mesenchymally”[All Fields] OR “mesenchymals”[All Fields] OR “mesenchymes”[All Fields] OR “mesoderm”[MeSH Terms] OR “mesoderm”[All Fields] OR “mesenchyme”[All Fields]))) AND (english[Filter]). All the studies included were screened for additional references.

Two of the authors (R.P and A.S) performed the searches between May and July 2021. Articles which were published between January 2010 and July 2021 were searched.

### 2.5. Data Collection Process

Two of the authors (R.P. and A.S.) worked independently on the data collection method. A third reviewer was brought in to resolve any disagreements (S.D.S). The following was the screening procedure: R.P. and A.S. began by reviewing the title and abstract, then moved on to the full-text version. The articles that were not rejected based on titles and abstracts were read in their entirety. In the case of disagreement, S.D.S. intermediated. The number of publications included and excluded was reported using the PRISMA flowchart [5].

### 2.6. Data Items

Data regarding primary author, year of publication, animal species, level of evidence, sample size, sex, checkpoints, type of induction and classification type (mechanical, surgical and chemical), and joint involved were extracted.

### 2.7. Risk of Bias

The Risk of Bias in Non-Randomized Studies of Interventions methodological index for non-randomized studies (MINORS) [6] by Cochrane was used to assess the possibility of bias in the included studies. Authors R.P. and A.S. separately scored the articles that were chosen. Any disagreements were resolved by the third reviewer, S.D.S.

## 3. Results

### 3.1. Study Selection

Overall, 1621 articles were found from the search of the relevant literature; however, after eliminating duplicates, only 1358 remained. Based on the title and abstract screening, 1298 of the 1358 papers were eliminated. A total of 60 articles were screened in full-text, and 24 were excluded due to a lack of data on population and accuracy (n = 5), incoherent conclusions on OA induction models (n = 5), a number of participants lower than 10 (n = 3) or unspecified data on population (n = 11). As a result, 36 articles passed the final screening and were included in the analysis. Figure 1 depicts the screening procedure.

### 3.2. Study Characteristics

The 36 selected articles included a total of 1434 animals, which were selected for the study. The final studies selected by the reviewers included the following level of evidence: Level IV 36 case-control studies. Eight of the studies included information about the chemical induction of OA [1,7,8,9,10,11,12,13], nineteen of the studies were focused on mechanical induction [4,14,15,16,17,18,19,20,21,22,23,24,25,26,27,28,29,30,31,32,33,34,35] and nine of the studies on induced OA through surgery [3,17,20,21,22,35,36,37,38]. None of the included studies adopted the genetic induction model; therefore, this item was not assessed. In addition, 35 studies specified the timing at which the animals were checked, making the longest follow-up at 20 weeks post-induction. All the study characteristics are summarized in Table 1 and Table 2.

### 3.3. Quality of Evidence

Using the *MINORS* tool, thirty-six studies were scored as having a “high risk of bias”. The studies examined were all similar in design, with several missing follow-ups [40] or failing to account for potential confounding areas among the variables. The RoB-2 tool was not used, as randomized clinical trials were not found. The *MINORS’* screening process is reported in Table 3.

### 3.4. Diagnostic Procedure

In twenty-three of the studies, the diagnostic procedure involved to evaluate the cartilage severity damage was the histological method [3,4,8,9,11,12,13,15,17,18,19,20,21,22,24,37,38,39]. Eleven studies selected the µCT scan as a radiological investigation method [4,7,8,13,15,21,23,27,28,29,31,32,33,37,38], either in combination with the histological test or individually, in order to diagnose the damage to the joint structures. In only two studies [15,37], magnetic resonance imaging (MRI) [41] was used for the diagnosis. The diagnostic timing varied across the studies considered. All the data are reported in Table 2.

### 3.5. Animal Models

In the 36 studies selected, several animal species were used for the OA evaluation process, with mice being the most utilized animal model. Fifteen [4,21,23,25,26,27,28,30,31,32,33,34,35,39,42] authors selected male and female C57BL/6N mice, CBA mice, Str/ort mice, FVB strain mice, LGXSM-6 and LGXSM-33 mice. C5BL/6N mice and CBA mice were selected as the most suitable species for the experiments. However, eighteen authors [1,3,4,7,8,9,10,11,12,13,14,15,17,18,19,22,24,36,37,38,39] preferred to study and induce OA in other animal species, such as Wistar Rats, Sprague Dawley rats, Lewis rats, Flemish Giant and New Zealand rabbits. Furthermore, two other authors [16,20] selected pigs and Suffolk-cross sheep, respectively. All the data are reported in Table 2.

### 3.6. Type of Induction

Of the eight studies that specified OA induction in animals utilizing a chemical approach, five were caused by an intra-articular injection of monoiodoacetate (MIA) [1,7,8,12,13]. In contrast, the other three models were determined by an injection of collagenase [11], a urinary plasminogen activator injection [9] and an injection of complete Freund’s adjuvant [10], respectively. The majority of the studies selected focused on the mechanical inductive approach through in vivo mechanical loadings. Conversely, of the nine studies focused on the surgical approach for the induction of OA in animals, four performed a surgical destabilization by ACLT [17,20,37,38], three performed a medial meniscus debridement (MMD) [3,21,35], one performed a medial collateral ligament tear (MCLT) [36] and one performed sham surgery [22]. All the data are reported in Table 2.

### 3.7. Type of Joint Involved

In the OA induction process, the authors focused on a specific joint in a specific animal model. As a consequence, the knee joint resulted as the most selected one, remarkably in murine animal models. For the knee joint, 16 studies preferred to induce OA in mice [4,15,21,25,26,27,28,29,30,31,32,34,35,39,43,44], 9 in rats [1,7,11,12,13,14,17,19,36,38], 5 in rabbits [3,18,22,24,37] and only 1 in sheep [20] and pigs [12], respectively. Subsequently, in the rat animal models, two papers elected the lumbar facet joints [18,19] and, similarly, OA was induced in the hip joint [13] by only one author. All the data are reported in Table 2.

## 4. Discussion

OA is considered to be one of the most significant diseases of the future due to the aging population [45,46,47].

Finding new solutions for studying this condition and testing drugs or surgical methods is crucial. Reporting the latest findings in the international literature can assist researchers worldwide in conducting more accurate studies using the most appropriate animal models for OA conditions. This paper has reviewed the latest literature to gather information about different animal models and induction methods for OA.

The reviewed studies utilized mechanical, surgical and chemical methods to induce OA. Most of the studies used mechanical induction, while surgical and chemical induction were performed less frequently. The studies primarily focused on the mechanical approach for inducing OA in animals using mechanical loading devices. In contrast, the surgery-induced approach focused on trauma caused by (micro)surgery, including ACLT, MMD, MCLT and MMT [18].

Diet, drugs, age, environmental pollution and many other factors have an impact on the synovial fluid’s composition and, therefore, its lubricating properties [48,49]. The major cause of OA, excluding mechanical or surgical injuries, are changes in the chemical composition of the synovial fluid. This is also the reason why there is contact between the articular surfaces; their abrasion and damage is identified as OA. Sulaiman et al. [50] compared the synovial fluid proteomics changes in surgical- and chemical-induced OA rabbit models. This study demonstrated that the surgically induced model showed a wider range of proteome profile; conversely, the chemically induced joints had a slower OA progression, showing inflammatory changes at the early phase but having a decreased expression at the later stages. Although several OA animal models have been reported, it is unknown at this time if any of them truly display these “ideal” characteristics. OA and other musculoskeletal diseases are frequently studied using mice, which are a practical model species [43].

### 4.1. Mechanical Models

Non-invasive mechanical mouse models of OA that do not require surgery or invasive techniques are a valuable advancement in studying the mechanisms and treatments of OA. Compared with invasive injury models, these non-invasive models initiate joint degradation in a more controlled and accurate manner, replicating human damage circumstances. Mechanical force is applied externally to the joint to cause harm to the bones, cartilage or soft tissues [43]. Some studies, such as that of Rai et al. [15], used a multi-cyclic loading pattern to cause significant cartilage fibrillation and degradation over a prolonged period. On the other hand, Christiansen and colleagues [23] used a single-cycle loading strategy that induced ACL rupture and resulted in mild OA over a longer period, with cartilage lesions observed in all the compartments of the knee joint. This approach released energy that likely reduced direct injury to the cartilage and indicates that joint destabilization, rather than direct cartilage injury, is caused by a single episode of tibial compression. In Rai et al.’s model [15], the loading pattern directly impacted the articular cartilage, resulting in a more accurate representation of human knee trauma. Both models suggest that human OA is usually the result of a series of loading events rather than a single traumatic experience.

### 4.2. Surgically Induced Models

Surgical damage techniques such as anterior cruciate ligament transection (ACLT) induce joint degradation by rupturing ligaments and menisci, altering the joint’s stability and biomechanics [13,43]. However, as noted by Mohan et al. [13], ACL tears often accompany acute meniscal damage, which is not accounted for in ACLT models. The meniscal transection (mACLT) model destabilizes the knee by transecting the ACL and partially transecting the meniscus. The tibiofemoral compressive impact model (ACLF) causes ACL rupture and injury to adjacent tissues, including the meniscus, through a single blunt-force hit to the tibiofemoral joint [13]. The surgically induced OA model (DMM) in mice, according to McNulty et al. [35], is highly reproducible, inducing moderate to severe articular cartilage lesions in young mice within 8 weeks post-surgery. Lorenz et al. [46] suggest considering gender-specific effects and evaluating data separately for males and females in mouse models, as sex hormones play a crucial role in OA development in DMM surgery models. Foxa et al. [47] described generating an OA-like model in mice through bilateral DMM surgery, with male mice being excellent subjects due to their increased likelihood of developing OA. The shift of the tibia anterior to the femur after DMM surgery, as noted by Moodie and colleagues [21], significantly increases the tibial AP range of motion, potentially damaging the ACL due to the proximity of the medial meniscal tibial ligament to the ACL’s tibial insertion [35].

### 4.3. Chemical Models

Chemical models of OA have a distinct pathophysiology and bear no resemblance to post-traumatic OA. They are also less invasive than surgical models [48]. The ease of induction and reproducibility make them ideal for short-term trials [48,49]. Pitcher et al. [49] reported that MIA models are responsive to conventional pain-relieving therapies, making it possible to test preventive and therapeutic procedures for OA pain management in compound development. However, Mohan et al. [13] noted that the gradual progression of OA in chemical induction models may be superior, but the data are ambiguous.

Currently, the most utilized type of chemical induction is MIA, which has an inhibitory activity of glyceraldehyde-3-phosphate dehydrogenase glycolysis and induces the death of chondrocytes, according to Morais et al. [7]. In both rodents and non-rodents, the intra-articular injection causes chondrocyte destruction. When administered in rats, it causes cartilage lesions with loss of proteoglycan matrix and functional alterations with stiffness that are analogous to those seen in human osteoarthritis [7]. The study by Miyamoto et al. [8] developed a rat hip OA model induced by an intra-articular injection of MIA. The histological appearance of MIA-injected hips showed an OA-like appearance. The current work describes the temporally dependent course of radiographic and subchondral bone alterations in the rat hip histology. It is very interesting to observe how osteoarthritis progresses over time following the MIA injection [8].

### 4.4. Animal Models Characteristics

McCoy et al. [51] suggested that, as with any species, only skeletally mature mice should be used in models of OA. Likewise, in rats, the recommended age for this species to participate in any study is 3 months. The earliest age at which it is recommended to include guinea pigs in any study is 6 months of age, by which time spontaneous lesions may already be established. Finally, in sheep and goats, it should be noted that skeletal maturity may not be reached until at least 2 years of age.

For the study of the pathogenesis and pathophysiology of the disease, Serra et al. [52] observed that small animals (mice, rats, guinea pigs and rabbits) are the most frequently utilized models, primarily because these models have a faster disease progression and are relatively cheap and simple to handle. Based on the results proposed by Ding et al. [53], the rat chondrocyte inflammation model may help in the study of the early pathological mechanism of OA, demonstrating considerable differences from mice. Therefore, various animal species have been found to be relevant to the OA evaluation process, with mice being the most used surgical and mechanical induction model. In contrast, rats are equally valuable to the latter for chemical induction.

Moreover, sheep and rabbits have also been considered acceptable for the mechanical method. Even though the heterogeneity of animal species appears as an important tool to evaluate different aspects of the osteoarthritic event, these models’ main drawbacks include their small size, which reduces the quantity of tissue available for biochemical studies, and the differences in tissue structure and joint mechanics between these species and humans. [51].

The fact that mice are much smaller than humans leads to noticeably different biomechanical loads on their joints, which cannot be overlooked. Additionally, the cartilage anatomy of mice differs noticeably from that of humans in, for instance, the overall cartilage thickness (on average, 30 mm; 50-fold thinner than in humans), a thick layer of calcified cartilage and a lack of distinct superficial, transitional and radial zones of chondrocytes [51]. Because rat articular cartilage is thicker (*0.1 mm) than mouse articular cartilage, whole and partial thickness lesions in rat cartilage can be created to examine therapeutic approaches for cartilage healing. However, the biomechanics of the rat’s stifle-joint differs significantly from that of the human joint, resulting in distinct patterns of cartilage stress [51].

Rabbits have been used for decades to induce OA both chemically and surgically despite the marked differences in joint biomechanics and gait when compared to humans. Rabbit stifles have a significantly higher flexion angle than that of humans. In addition, rabbit cartilage is 10 times thinner than human cartilage (0.3–0.7 vs. 2–3 mm) but has a much higher chondrocyte density [51]. Sheep and pigs represent an advantageous model due to their larger size, which means easier maneuverability, amenability to diagnostic imaging, effortless synovial fluid collection, arthroscopic intervention and postoperative management.

Sheep are favored as a large animal species due to the marked similarity between their stifle anatomy and that of the human knee. Pigs, on the other hand, differ somewhat in size with respect to the cruciate ligament and width of the meniscus, but the remarkable similarities with humans in gastrointestinal physiology and the immune system make this species an attractive option for testing pharmaceutical and biological interventions [51].

Throughout this study, it has been highlighted that sixteen authors preferred to induce OA in knee joints predominantly in mice, ten in rats, five in rabbits and only one in sheep and pigs, respectively. Moreover, two studies selected lumbar facet joints in rats, and only one paper chose hip joints in rats. The knee joints and lumbar facet joints resulted in the most used anatomical models. Shuang et al. [10] highlighted them as the most prone to destabilization to induce OA. Moreover, there is only one author [8] who preferred to use the hip joint to induce OA.

### 4.5. Limitations

The differences between studies in terms of follow-up, animal model and type of induction made it impossible to perform a meta-analysis. While numerous animal models of OA have been described, no consensus exists on the injury methods used to induce OA development, since joint degeneration likely depends on the injury method used. Moreover, various scales have been used to describe the settlement of the osteoarthritic event, showing a preference for the Osteoarthritis Research Society International (OARSI) histopathology grading and the Mankin systems [54]. The OARSI scale assigns a grade of damage from 0 to 6, based on the depth of OA advancement into the cartilage, and a stage of damage from 0 to 4, based on cartilage involvement. The combined value of grade and stage (score range 0–24) determines the final score. On the other hand, the Mankin scoring system has been established for grading OA changes. According to Miyamoto et al., Mankin [8] ratings are based on five factors [7]. These include shape, chondrocyte quantity, chondrocyte clustering, proteoglycan content, subchondral bone plate and/or tidemark modifications, and a vascular invasion into cartilage. Low (1–3 points), moderate (4–7 points) and high (>8 points) total points were assigned. Due to an inhomogeneous or lacking ranking scale, it has been found difficult to classify the most suitable animal model for OA evaluation. Moreover, OA development is not a linear process. As reported in the article published by Longo et al. [55] in 2021, the type of animal model used and the circumstances in the stables and cages may have an impact on the experiment outcomes.

The animal model of OA could be spontaneous or induced; however, the topic of the current review is focused on induced models. Spontaneous models of OA include a wide number of articles and topics of discussion; therefore, these models were not included as they deserve a dedicated article. Further reviews are required to report the most recent findings on spontaneous models.

All the articles included reported a high risk of bias according to the MINORS scale. Lastly, it was not possible to perform a meta-analysis due to the high heterogeneity between the studies.

## 5. Conclusions

In the field of OA research, animal models are commonly used, but a universally accepted method has yet to be established. The varying OA grading scales make it challenging to determine the severity of OA. While mice are frequently used in studies, there is still a significant diversity in the choice of animals. Despite this, several induction models, including mechanical, surgical and chemical, have been found to be effective in inducing OA in animals, with the mechanical model being the most popular and consistent with the OA process, the surgical model being the most invasive, and the chemical model being the most time-dependent. To better understand OA development, further high-quality studies with consistent sample sizes and classification methods are needed to identify the most valid and reliable animal model.

## Figures and Tables

**Figure 1 biology-12-00283-f001:**
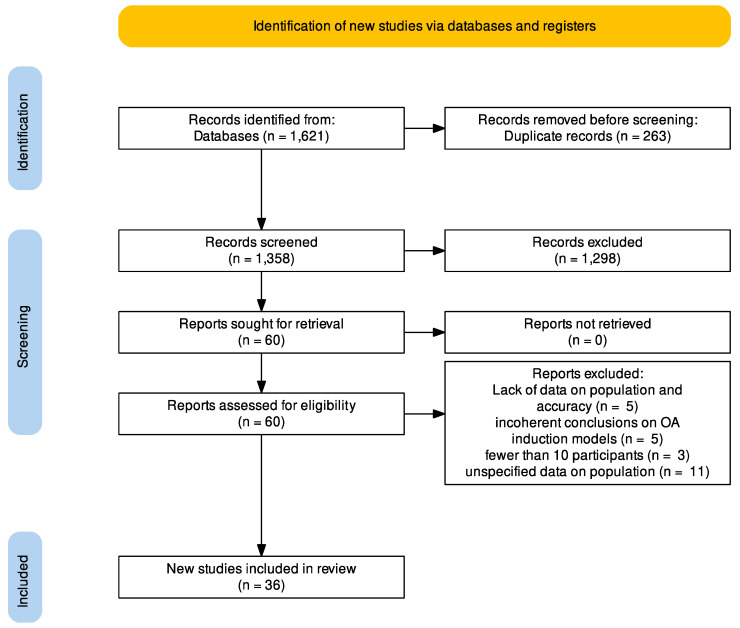
Prisma flowchart.

**Table 1 biology-12-00283-t001:** Study characteristics.

Author	Level of Evidence	Year	Animal	Sex	N of Samples	Induction	Type of Induction	Joint Involved	Endpoint
Adães et al. [11]	LOE 4	2014	Wistar rats	M	72 (n = 24)	C	Two IAI of collagenase	Knee	1, 2, 3, 4, 5, 6 wk post-inj.
Allen, Kyle D et al. [36]	LOE 4	2012	Lewis rats	U	16 (n = 3)	S	MCL was transected	Knee	9-24-28 d
Arunakul et al. [3]	LOE 4	2013	NZW rabbits	M	26 (n = 5)	S	MMD	Knee	8 wk
Christiansen et al. [23]	LOE 4	2012	C57BL/6N mice	M	48	M	Single overload cycle of tibial compression	Knee	1, 3, 7, 14, 28, or 56 d
Etienne J O O’Brien et al. [20]	LOE 4	2012	SC sheep	F	29 (n = 5, n = 7)	S	Transection of ACL with consequent reconstruction	Knee	0-4-20 wk
Farooq Rai et al. [15]	LOE 4	2017	LGXSM-6 and LGXSM-33 mice	M	88 (n = 44)	M	Axial tibial compression	Knee	5, 9, 14, 28, 56 d
Fischenich et al. [37]	LOE 4	2016	FG rabbits	M and F	33	S	ACLT, ACLF and mACLT	Knee	4, 8, 12 wk
Geetha Mohan et al. [13]	LOE 4	2011	Wistar rats	M	12	C	Injection of MIA	Knee	0-2-6-10 wk
Horisberger et al. [18]	LOE 4	2013	NZW rabbits	M	24 (n = 8)	M	Joint loading by muscle stimulation	Knee	NS
Joana Ferreira-Gomes et al. [12]	LOE 4	2012	Wistar rats	M	54 (n = 5)	C	Injection of MIA	Knee	3-7-14-21-31 d after MIA injection
Killian et al. [24]	LOE 4	2010	FG rabbits	U	11	M	Tibiofemoral impaction resulting in ACL rupture or surgical ACL transection	Knee	12 wk for traumatically torn and after 12 wk in ACL transected animals
Ko et al. [25]	LOE 4	2013	C57BL/6 mice	M	63	M	Cyclic compression	Knee	1, 2 and 6 wk
Ko et al. [26]	LOE 4	2016	C57Bl/6 mice	M	21	M	VCL	Knee	0, 1 and 2 wk
Korostynski et al. [1]	LOE 4	2018	Wistar rats	M	32 (n = 8)	C	IAI of MIA	Knee	2, 14, 28 d
Lewis et al. [27]	LOE 4	2011	C57BL/6 mice	M	50 (n = 25)	M	Low and high energy fractures after loading	Knee	0, 1, 3, 5 and 7 d post-fracture
Lynch et al. [28]	LOE 4	2010	C57BL/6 mice	M and F	n = 14/sex	M	Dynamic compression of the left tibia	Knee	2 wk of load evaluation
Lockwood et al. [29]	LOE 4	2013	C57BL/6N mice	M	80	M	Tibial compressive overload	Knee	0 d, 10 d, 12 wk or 16 wk
Maerz et al. [38]	LOE 4	2016	Lewis rats	F	36 (n = 6)	S	ACL rupture or ACL transection	Knee	4 or 10 wk
McCulloch et al. [16]	LOE 4	2014	Pig	U	36	M	Shear loading model	Knee	3, 7, 14 d
McNulty et al. [35]	LOE 4	2012	C57BL/6 mice	M	11	S	DDM surgery, Sham surgery	Knee	2 months
Miyamoto et al. [8]	LOE 4	2016	SD rats	M	60 (n = 30)	C	IAI of MIA	Hip	7, 14, 28, 42, 56 d
M.L. Roemhildt et al. [19]	LOE 4	2012	SD rats	M	25 (n = 5)	M	VLD implement	Knee	0-6-20 wk
Moodie, J.P. et al. [21]	LOE 4	2011	C57Bl6 mice	M	38	S	DMM surgery	Knee	0-4-8 wk
Morais SV et al. [7]	LOE 4	2016	Wistar rats	M	48 (n = 24)	C	IAI of MIA	Knee	1, 7, 14, 21, 28 d
Nomura et al. [4]	LOE 4	2017	C57BL/6J mice	M	24 (12 × 2; n = 4)	M	Hindlimb unloading	Knee	2, 4, 8 wk
Onur et al. [30]	LOE 4	2013	FVB strain mice	U	21	M	Axial compression	Knee	1, 8 wk
Poulet et al. [31]	LOE 4	2011	CBA mice	U	35	M	Cyclic loading compression	Knee	2, 3 and 5 wk
Poulet et al. [32]	LOE 4	2013	Str/ort mice	M	n = 3, n = 6, n = 8	M	Compression with servo-hydraulic materials	Knee	2 wk
Poulet et al. [33]	LOE 4	2015	CBA mice	M	n = 8, n = 8, n = 5	M	Compression with servo-hydraulic materials	Knee	10, 13 wk
Qin et al. [17]	LOE 4	2014	SD rats	F	88	S	ACLT	Knee	6, 12, 18 wk
Ramme et al. [14]	LOE 4	2017	SD rats	F	24	M	ACL biomechanical rupture and surgical transection	Knee	U
Satkunananthan et al. [39]	LOE 4	2014	C57BL/6 mice	M and F	54	M	ACL rupture induced by tibial compression	Knee	1-7 d
Shuang et al. [10]	LOE 4	2014	SD rats	M	60	C	IAI of complete Freund’s adjuvant [9]	Spine	3, 7, 14, 21 and 28 d
Shuang et al. [9]	LOE 4	2015	SD rats	M	n = 88	C	IAI of uPA [9]	Spine	3, 7, 14, 28, 42, 56 d
Vaseenon, Tanawa et al. [22]	LOE 4	2011	NZW rabbits	U	40	S	Sham surgery	Knee	0-8-16 wk
Wu et al. [34]	LOE 4	2014	C57BL/6 mice	M	n = 54	M	Compressive joint loading	Knee	5, 9, 14 d

**Table 2 biology-12-00283-t002:** Type of induction and joint involved.

Author	OA Score	Conclusions
Adães et al. [11]	Knee-bend and catwalk	It induces significant nociceptive alterations associated with OA changes
Allen, Kyle D et al. [36]	OARSI score present	First description of functional losses after medial meniscus injury in the rat with reported changes in gait dynamics
Arunakul et al. [3]	Mankin score	Results support the MMD technique as being an effective surgical insult modality to replicate injurious cumulative abnormal cartilage loading
Christiansen et al. [23]	OARSI scale	This model is a significant improvement over other mouse models of PTOA, since it induces an injury that is translatable to humans, easy to implement and highly reproducible.
Etienne J O O’Brien et al. [20]	Hellio Le Graverand protocol	Shows how a reconstruction model is not really helpful
Farooq Rai et al. [15]		High mechanical loading instigated whole knee-joint changes leading to PTOA
Fischenich et al. [37]		Dependency of the model on the location, type and progression of damage over time
Geetha Mohan et al. [13]	OARSI score = 17	Low-dose MIA-induced OA rat model mimics the human disease condition and clearly demonstrates disease progression in the tibial subchondral bone in a timely manner
Horisberger et al. [18]	Mankin score	Muscular loading of physiological magnitude but excessive intensity caused chondrocyte deaths and onset of early OA in rabbit knees
Joana Ferreira-Gomes et al. [12]	Knee-bend and catwalk	Concentrated more on the neuronal aspect rather than the causes of OA, suggests that axonal injury and a regeneration response may be happening in this model of OA
Killian et al. [24]		This study has implications for the future use of lapine models for osteoarthritis, as it incorporates traumatic loading as a more realistic progression of osteoarthritis compared with surgically transected models
Ko et al. [25]	OARSI scale	Noninvasive loading model, permits dissection of temporal and topographic changes in cartilage and bone
Ko et al. [26]	Osteomeasure histomorphometry system, OsteoMetrics, Decatur, GA	Demonstrate that a single session of noninvasive loading leads to the development of OA
Korostynski et al. [1]		The progression of cartilage damage is driven by the complex but precise regulation of gene patterns
Lewis et al. [27]	Mankin score	This study demonstrates that articular fracture is associated with a loss of chondrocyte viability and increased levels of systemic biomarkers
Lynch et al. [28]		For all cancellous measures, the response to tibial compression did not differ between male and female mice
Lockwood et al. [29]	OARSI score	These studies further characterize the non-invasive knee-injury mouse model
Maerz et al. [38]	Mankin score	AC degeneration is a time-, compartment- and injury-dependent cascade
McCulloch et al. [16]		The chondrocytes in the shear specimens are attempting repairs but are unable to mount a successful effort
McNulty et al. [35]	Mankin score	Provided surgically induced models using a comprehensive histological grading scheme
Miyamoto et al. [8]	Mankin score	Intra-articular injection of MIA consistently causes progressive hip OA
M.L. Roemhildt et al. [19]		Shows VLD is a better replicate of how OA really develops
Moodie et al. [21]	Graded but without scaling grade	Increase in AP laxity suggests that DMM surgery redistributes loading posteriorly on the medial plateau, resulting in bone and cartilage loss
Morais SV et al. [7]		OA induced by intra-articular MIA is a good model to be used in related research
Nomura et al. [4]		Thinning of articular cartilage induced by mechanical unloading may be mediated by metabolic changes in chondrocytes
Onur et al. [30]	Pritzker grading scale	Axial compression without joint instability does not appear to be sufficient for inducing PTOA
Poulet et al. [31]		This model offers opportunities to study the effects of various loading magnitudes and regimens on joint health and disease
Poulet et al. [32]		Load application appears to accelerate OA
Poulet et al. [33]		Concomitant AC-damage aggravates focal SCB-thickening induced by applied loading
Qin et al. [17]	OARSI score	LMHFV accelerated cartilage degeneration and deterioration of OA and promoted bone formation in affected distal femur epiphysis
Ramme et al. [14]		ACL reconstruction and joint assessment will be useful in future joint-injury/PTOA studies in small-animal models
Satkunananthan et al. [39]	OARSI scale	Describes the dynamic protease profile following traumatic knee injury and establishes FRI as a useful analysis method
Shuang et al. [10]	OARSI score	Injection of complete Freund’s adjuvant caused local synovitis in early stage, and the inflammation was similar with osteoarthritis
Shuang et al. [9]	OARSI score	This animal model is convenient and shows good resemblance of human facet joint OA pathology
Vaseenon, Tanawa et al. [22]	Mankin score	The accompanying localized incongruity involved the onset of biomechanical abnormality consistent with causing cartilage degeneration
Wu et al. [34]		Established a murine model of knee-joint trauma with different degrees of overloading in vivo

**Table 3 biology-12-00283-t003:** MINORS.

Author	Clearly Stated Aim	Inclusion of Consecutive Patients	Prospective Data Collection	Checkpoints Appropriate to Study Aim	Unbiased Assessment of Study Endpoint	Follow-Up Period	Appropriate to <5% Lost to Follow-Up	Prospective Calculation of Study Size	Total Score
Adães, Mendonça et al., 2014 [11]	2	2	2	2	0	0	0	2	**10**
Allen, Mata et al., 2012 [36]	2	2	2	2	0	0	0	2	**10**
Arunakul, Tochigi et al., 2013 [3]	2	2	2	2	0	0	0	2	**10**
Christiansen et al., 2012 [23]	2	2	2	2	0	0	0	2	**10**
Fischenich, Pauly et al., 2017 [37]	2	2	2	2	0	2	0	2	**12**
Mohan, Perilli et al., 2011 [13]	2	2	2	2	0	0	0	2	**10**
Horisberger, Fortuna et al., 2013 [18]	2	2	2	0	0	0	0	2	**8**
Ferreira-Gomes, Adães et al., 2012 [12]	2	2	2	2	0	0	0	2	**10**
Killian et al., 2010 [24]	2	2	2	2	0	0	0	2	**10**
Ko et al., 2013 [25]	2	2	2	2	0	0	0	2	**10**
Ko et al., 2016 [26]	2	2	2	2	0	0	0	2	**10**
Korostynski, Malek et al., 2018 [1]	2	2	2	2	0	0	0	2	**10**
Lewis et al., 2011 [27]	2	2	2	2	0	0	0	2	**10**
Lynch et al., 2010 [28]	2	2	2	2	0	0	0	2	**10**
Lockwook et al., 2014 [29]	2	2	2	2	0	0	0	2	**10**
Maerz et al., 2016 [38]	2	2	2	2	0	0	0	2	**10**
McCulloch et al., 2014 [16]	2	2	2	2	0	0	0	2	**10**
McNulty et al., 2012 [35]	2	2	2	2	0	0	0	2	**10**
Miyamoto, Nakamura et al., 2016 [8]	2	2	2	1	0	0	0	2	**9**
Moodie et al., 2011 [21]	2	2	2	0	0	0	0	2	**8**
Morais, Czeczko et al., 2016 [7]	2	2	2	2	0	0	0	2	**10**
Nomura, Sakitani et al., 2017 [4]	2	2	2	2	0	0	0	2	**10**
O’Brien et al., 2012 [20]	2	2	2	2	0	0	0	2	**10**
Onur et al., 2013 [30]	2	2	2	2	0	0	0	2	**10**
Poulet et al., 2011 [31]	2	2	2	2	0	0	0	2	**10**
Poulet et al. 2013 [32]	2	2	2	2	0	0	0	2	**10**
Poulet et al., 2015 [33]	2	2	2	2	0	0	0	2	**10**
Qin, Chow et al., 2014 [17]	2	2	2	2	0	0	0	2	**10**
Ramme, Lendhey et al., 2018 [14]	2	2	2	0	0	0	0	2	**8**
Rai et al. 2017 [15]	2	2	2	0	0	0	0	2	**10**
Roemhildt, Beynnon et al., 2013 [19]	2	2	2	2	0	0	0	2	**10**
Satkunananthan et al., 2014 [39]	2	2	2	2	0	0	0	2	**10**
Shuang, Hou et al., 2015 [9]	2	2	2	2	0	0	0	2	**10**
Shuang, Zhu et al., 2014 [10]	2	2	2	2	0	0	0	2	**10**
Vaseenon, Tochigi et al., 2011 [22]	2	2	2	2	0	0	0	2	**10**
Wu et al., 2014 [34]	2	2	2	2	0	0	0	2	**10**

## Data Availability

The datasets generated and/or analyzed during the current study are not publicly available due to analyses being underway for subsequent publications. However, they are available from the corresponding author on reasonable request.

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
