# Peer review of "Induced Models of Osteoarthritis in Animal Models: A Systematic Review"

_biology, 2023, doi:10.3390/biology12020283_

Round 1
Reviewer 1 Report
The authors of the manuscript entitled: "Induced models of Osteoarthritis in Animal Models: A Systematic Review" present the results of the analysis of 36 papers, published in the period 2010-2021, selected from 1621, which present the results of research on models of osteoarthritis induction in laboratory animals. In total, the results include studies performed on 1,472 animals. From a statistical point of view, this number seems to be sufficient. This is where the first imperfections in the selection of articles appear. First, we have many different species of animals here: mice, rats (in this case they are biologically similar), rabbits, pigs and sheep. Secondly, the studies cover different joints. Are the authors able to show, for example, how many papers concerned OA modeling of one type of joint in one group of animals? I'm afraid this will be data from only a few papers published by one research group that focused on a specific joint in a specific animal species. In this case, we cannot talk about a statistical sample.
Mechanical injuries or surgical intervention are not particularly interesting in this case. Any intervention in the structure of the joint violates its integrity and never comes to a complete cure. The rate of development of the degenerative disease is only a matter of time and the individual predisposition of the patient. The authors should focus on the third method, i.e. chemical induction. This method seems to be the most interesting. There are many examples of OA in the literature in patients who have never suffered joint injuries. The synovial fluid is very complex from a chemical point of view. There are many factors (diet, medications, age, environmental pollution, and many others) that affect the composition and thus the lubricating properties of the synovial fluid. Excluding the typical mechanical injuries and surgical interventions, it is the disturbances in the chemical composition of the synovial fluid that are the primary factor leading to OA. Impaired lubrication of articular cartilage causes its mechanical damage through mutual abrasion of articular surfaces. In addition to insufficient lubrication, we are dealing with a reduced ability of the synovial fluid to absorb shocks and transfer loads. This is also the reason why there is contact between the articular surfaces and their abrasion and damage, which is identified as OA.
Thus, the authors should go in the direction of chemical models of OA and do a similar analysis for one group of animals and one type of joint.
It is not the fault of the authors of the manuscript that research into OA induction models is not being conducted in the right direction. The presented analysis only confirms the reader in this belief. However, research should bring something new, and here it does not show.
If the Authors believe otherwise, please indicate this in the revised/supplemented version of the manuscript.
Author Response
Dear reviewers,
We would like to thank you for the helpful comments and suggestions. We have revised the paper accordingly and hope that the work is now ready for publication. The changes are itemized below with our comments (dark blue text) to the reviewer’s suggestions. Changes made in the text are highlighted in yellow in the original manuscript.
Reviewer’s comments
Reviewer #1
The authors of the manuscript entitled: "Induced models of Osteoarthritis in Animal Models: A Systematic Review" present the results of the analysis of 36 papers, published in the period 2010-2021, selected from 1621, which present the results of research on models of osteoarthritis induction in laboratory animals. In total, the results include studies performed on 1,472 animals. From a statistical point of view, this number seems to be sufficient. This is where the first imperfections in the selection of articles appear.
First, we have many different species of animals here: mice, rats (in this case they are biologically similar), rabbits, pigs and sheep.
Thanks for your comment. According to our selection criteria, we decided to include almost all animal species described in literature involved in the induction of OA to provide a full inclusive systematic review on this topic . As a consequence, we focused our research on the most common types of OA induction methods in several animal models. Even if many species have been included, the majority of the articles selected mice as the most privileged OA animal model. We agree with the reviewer that rats are biologically similar to mice, however these two species present some differences in the early pathological mechanism of OA, according to several authors cited in our paper. Thanks to your comment we improved our paper, and we clarified this point in the discussion section.
Secondly, the studies cover different joints. Are the authors able to show, for example, how many papers concerned OA modeling of one type of joint in one group of animals?
I'm afraid this will be data from only a few papers published by one research group that focused on a specific joint in a specific animal species. In this case, we cannot talk about a statistical sample.
Thanks for your comment. In literature the knee joint resulted as the most used anatomical model for the induction of OA in animals. In our research, as reported in table 2, the knee joint resulted again the most selected joint in mice (16), rats (10), rabbits (5) sheep (1) and pig (1) followed by lumbar-facets joint (2) and hip joint (1) in rats. Due to our availability of these kind of data, we decided, as suggested, to improve the discussion section quantifying how many papers focused on a specific joint in a specific animal model. We improved the result section accordingly.
Mechanical injuries or surgical intervention are not particularly interesting in this case. Any intervention in the structure of the joint violates its integrity and never comes to a complete cure. The rate of development of the degenerative disease is only a matter of time and the individual predisposition of the patient.
We would thank you for the possibility to clarify this point better. The aim of this research is to set up a comprehensive selection process which included all the papers present in literature regarding all the available methods able to induce OA in animals: mechanical, surgical and chemical. Therefore, we decided to make a large-scale analysis as to review and compare different OA animal models proposed by the selected authors. Mechanical models resulted as the most selected induction methods in animals since, especially the non-invasive mechanical mouse models which are a step-forward in OA research since they aseptically initiate joint degeneration and are more representative of human injury conditions than surgical or invasive injury models. Surgical models as well introduced very innovative procedures such as DMM surgery which is highly reproducible and changes induced by DMM surgery are strictly related with naturally occurring OA in aged joints.
The authors should focus on the third method, i.e. chemical induction. This method seems to be the most interesting.
Thanks for your comment. The most common OA induction methods founded in our review were mechanical surgical and chemical, however there is not a gold standard in the choice of the technique because they are all helpful in different contexts. For this reason, we decided to include all of the three most used methods described in literature. Despite that, we agree with the reviewer on the fact that the chemical one seems to be the most fascinating method, since it is the most time-dependent one allowing a gradual growing level of OA development. Therefore, thanks to your suggestions, we improved the discussion section focusing on the latter.
There are many examples of OA in the literature in patients who have never suffered joint injuries. The synovial fluid is very complex from a chemical point of view. There are many factors (diet, medications, age, environmental pollution, and many others) that affect the composition and thus the lubricating properties of the synovial fluid.
Excluding the typical mechanical injuries and surgical interventions, it is the disturbances in the chemical composition of the synovial fluid that are the primary factor leading to OA. Impaired lubrication of articular cartilage causes its mechanical damage through mutual abrasion of articular surfaces. In addition to insufficient lubrication, we are dealing with a reduced ability of the synovial fluid to absorb shocks and transfer loads. This is also the reason why there is contact between the articular surfaces and their abrasion and damage, which is identified as OA.
Thank you for the suggestion. The synovial fluid is a relevant parameter that should assessed in the analysis of OA establishment and progression. It seems that the surgical induced model showed a wider range of synovial proteome profile, whereas the chemical model showed lower OA progression expressing inflammatory changes at the early phase but having decreased expression at the later stages. Therefore, as indicated, we expand the discussion section focusing on the importance of synovial fluid characteristics.
Thus, the authors should go in the direction of chemical models of OA and do a similar analysis for one group of animals and one type of joint.
Thanks for the note. As previously reported, the discussion section has been revised and focused much more on chemical induction method.
It is not the fault of the authors of the manuscript that research into OA induction models is not being conducted in the right direction. The presented analysis only confirms the reader in this belief. However, research should bring something new, and here it does not show.
If the Authors believe otherwise, please indicate this in the revised/supplemented version of the manuscript.
Thank you for your advice, we believe that research should bring something new to the scientific community. In our study we tried to provide an updated and broad review of OA induction methods in animal models described in literature. The purpose of this article is to review recent literature on OA animal models to account for its use in future studies and understand the current procedural trends around the world. Still, not many studies value the importance of highlighting similarities and discrepancies between different aspects of OA induced in animal models, comparing several animal species, induction methods and joint selection. Therefore, as a systematic review, the aim of our paper is to present to the reader a reliable source of data regarding the above-mentioned criteria. In conclusion, since there is not an international consensus on the gold standard OA animal model, our hope is to start a general research trend on this topic to draw conclusions about it. Lastly, it was not possible to perform a metanalysis due to the high heterogeneity between studies. We improved the limitation section according to your suggestions
Reviewer 2 Report
The authors reviewed the current and prospective landscape of OA models in animal models. The manuscript is good for the journal. Before the manuscript can be published, the authors should address the following issues.
Authors should try to select more recent literature for reference. A relatively new, detailed approach to DMM by Gabrielle E Foxa et al. should be cited.
https://www.ncbi.nlm.nih.gov/pmc/articles/PMC7934896/. They also included some minor modifications for DMM, and recommend the male mice are more suitable for DMM.
Regarding the use of different animal species, the authors should have provided more details on the differences and applicability of the different animal species.
OA is a degenerative disease closely related to aging, and the authors should add and discuss the aging OA animal models.
Author Response
Dear reviewer,
We would like to thank you for the helpful comments and suggestions. We have revised the paper accordingly and hope that the work is now ready for publication. The changes are itemized below with our comments (dark blue text) to the reviewer’s suggestions. Changes made in the text are highlighted in yellow in the original manuscript.
Reviewer’s comments
Reviewer 2
The authors reviewed the current and prospective landscape of OA models in animal models. The manuscript is good for the journal. Before the manuscript can be published, the authors should address the following issues.
Thanks for the comment. We are honored you appreciated our paper.
Authors should try to select more recent literature for reference. A relatively new, detailed approach to DMM by Gabrielle E Foxa et al. should be cited. https://www.ncbi.nlm.nih.gov/pmc/articles/PMC7934896/. They also included some minor modifications for DMM, and recommend the male mice are more suitable for DMM.
Thanks to your advice, we updated and improved the most recent review on this topic, which helped us improve our paper's quality. As suggested, we considered to include in the research the article suggested by the reviewer. The discussion section has been enriched with these considerations gathering information from the previously mentioned article by Foxa et al. and the paper published by Lorenz et al. which is on the same line.
Regarding the use of different animal species, the authors should have provided more details on the differences and applicability of the different animal species.
Thank you for the note. In the studies selected, several animal species have been used for the evaluation process of OA, underlining that different animals are helpful in different contexts. Mice resulted as the most used animal model, however there is still a significant heterogeneity of articles that choose other animals. As suggested, we included in the discussion section some considerations from the article published by McCoy et al. which provide a complete comparison between small and large OA animal models analyzing disadvantages and dissimilarities between several animal species.
OA is a degenerative disease closely related to aging, and the authors should add and discuss the aging OA animal models.
Thank you for your advice. We agree with your comment, thus we decided to discuss the aging OA animal models including some considerations from the above mentioned articled published by McCoy et al. in which only skeletally mature animal species should be considered as OA animal models.